Are caves enough to represent karst groundwater biodiversity? Insights from geospatial analyses applied to European obligate groundwater-dwelling copepods

Galmarini Emma 1
Di Cicco Mattia 1
Fiasca Barbara 1
Mori Nataša 2
Iannella Mattia 1
Di Lorenzo Tiziana 3 4 5 6
Cerasoli Francesco 1
Galassi Diana Maria Paola dianamariapaola.galassi@univaq.it 1
1 Department of Life, Health & Environmental Sciences, University of L’Aquila , L’Aquila , Italy
2 Department for Organisms and Ecosystem Research, National Institute of Biology , Ljubljana , Slovenia
3 Research Institute on Terrestrial Ecosystems (IRET-CNR) , Florence , Italy
4 National Biodiversity Future Center (NBFC) , Palermo , Italy
5 Centre for Ecology, Evolution and Environmental Changes & CHANGE–Global Change and Sustainability Institute, and Departamento de Biologia Animal, Faculdade de Ciências, Universidade de Lisboa , Lisbon , Portugal
6 “Emil Racovită” Institute of Speleology , Cluj-Napoca , Romania
Caravaggi Anthony
Electronic publication date: 2025 Nov 6
Publication date: 2025
Volume: 13
Electronic Location ID: e20285
Received 2025 Jul 24; Accepted 2025 Oct 2
Copyright: ©2025 Galmarini et al.
Copyright year: 2025
Copyright holder: Galmarini et al.
License: This is an open access article distributed under the terms of the Creative Commons Attribution License, which permits unrestricted use, distribution, reproduction and adaptation in any medium and for any purpose provided that it is properly attributed. For attribution, the original author(s), title, publication source (PeerJ) and either DOI or URL of the article must be cited.
License URL: https://creativecommons.org/licenses/by/4.0/

Keywords: Copepoda, Groundwater, Biodiversity hotspot, Karst, Cave, Europe

Funding: Biodiversa+, the European Biodiversity Partnership under the 2021–2022 BiodivProtect European Commission GA No 101052342 Ministry of Universities and Research (Italy) Agencia Estatal de Investigación–Fundación Biodiversidad (Spain) Fundo Regional para a Ciência e Tecnologia (Azores, Portugal) Suomen Akatemia–Ministry of the Environment (Finland) Belgian Science Policy Office (Belgium), Agence Nationale de la Recherche (France) Deutsche Forschungsgemeinschaft e.V. (Germany) Schweizerischer Nationalfonds Grant No 31BD30_209583, Switzerland Fonds zur Förderung der Wissenschaftlichen Forschung (Austria) Ministry of Higher Education, Science and Innovation (Slovenia) Executive Agency for Higher Education, Research, Development and Innovation Funding (Romania)-Project DarCo This research was funded by Biodiversa+, the European Biodiversity Partnership under the 2021–2022 BiodivProtect joint call for research proposals, co-funded by the European Commission (GA No 101052342) and with the funding organisations Ministry of Universities and Research (Italy), Agencia Estatal de Investigación–Fundación Biodiversidad (Spain), Fundo Regional para a Ciência e Tecnologia (Azores, Portugal), Suomen Akatemia–Ministry of the Environment (Finland), Belgian Science Policy Office (Belgium), Agence Nationale de la Recherche (France), Deutsche Forschungsgemeinschaft e.V. (Germany), Schweizerischer Nationalfonds (Grant No 31BD30_209583, Switzerland), Fonds zur Förderung der Wissenschaftlichen Forschung (Austria), Ministry of Higher Education, Science and Innovation (Slovenia), and the Executive Agency for Higher Education, Research, Development and Innovation Funding (Romania)-Project DarCo. The funders had no role in study design, data collection and analysis, decision to publish, or preparation of the manuscript.

==============================
Caves are recognized as biodiversity hotspots for groundwater fauna, including obligate groundwater-dwelling copepods (Crustacea: Copepoda), exhibiting high species richness, endemism, and phylogenetic rarity. However, the extent to which caves alone provide a representative estimate of copepod species richness in karst areas remains uncertain. Taking advantage of the recently published EGCop dataset, the first expert-validated, Europe-wide occurrence dataset for obligate groundwater-dwelling copepods (hereinafter, GW copepods), this study investigates the distribution of GW copepods into karst areas, comparing species richness in caves versus other karst groundwater habitats (e.g., springs, karst streams, artificial wells), within and among the European karst units. The main aims are: (i) identifying karst areas which represent hotpots of GW copepod species richness; (ii) assessing to which extent caves, as open windows to the subterranean environments, contribute to define hotspots of GW copepods’ species richness into karst areas across Europe. EGCop comprises 6,986 records from 588 copepod species/subspecies distributed among four orders: Cyclopoida (3,664 records, 184 species), Harpacticoida (3,288 records, 395 species), Calanoida (32 records, seven species), and Gelyelloida (two records, two species). To perform geospatial analyses, we filtered the dataset by: (i) selecting only the records with spatial uncertainty in the associated coordinates lower than 10 km; (ii) searching for those records falling within, or very close to, the polygons representing European karst areas. Species richness hotspots were then estimated through geospatial analyses in geographic information system (GIS) environment. Within the selected records, those specifically referring to karst habitats (2,526 records, 369 species) are primarily represented by Harpacticoida (1,199 records, 228 species) and Cyclopoida (1,293 records, 132 species). Among species collected from karst habitats, records from caves (1,867, 73.9%) belong to 318 species (Harpacticoida = 189, Cyclopoida = 122, Calanoida = 7), representing 86.1% of the total species richness of karst habitats. Geospatial analyses reveal that the European hotspots of GW copepods’ species richness recorded exclusively in caves reflect the spatial arrangement of postglacial refugia in southern karst regions, though representing a subset of the broader diversity found across all karst groundwater habitats. Our findings highlight that the contribution of cave systems in groundwater biodiversity assessments and related conservation planning may vary depending on the evolution and morphologies of the target karst regions—often pointing to a high representativeness of caves for subterranean biodiversity, sometimes revealing their lower explanatory power within the broader karst systems.

Introduction

Karst landscapes are natural systems that develop on soluble rocks such as limestone, dolostone, and evaporites, with their formation largely attributed to chemical dissolution processes. Karst terrains cover approximately 10% of the Earth’s surface, provide fresh drinking water to an estimated 10% of the global population, and, according to some estimates, supply up to 25% of the groundwater used for agricultural and industrial purposes (Kuniansky et al., 2022). The classic karst landforms, which include enclosed depressions, sinking streams, and caves, are primarily generated by surface and subsurface waters dissolving rocks, mechanical erosion playing a secondary role (Andreychouk, 2016; De Waele, 2017; Zerga, 2024). Caves have always been considered the iconic elements of the karst. In fact, when we talk about karst biodiversity, we almost always refer to the biodiversity found in cave environments (Culver & Pipan, 2013; Moldovan, Kováč & Halse, 2018; Ponta & Onac, 2019; Culver & Pipan, 2019; Culver et al., 2021; Deharveng et al., 2024). This assumption finds a persuasive argument in the fact that caves are the windows open onto karst systems, they are normally more accessible to humans, and can be easily inspected. Indeed, research into subterranean biodiversity largely originated in caves. That many caves are biodiversity hotspots is undeniable. On the other hand, for many karst areas other environmental typologies are known such as basal karst springs, surface streams fed by karst waters, which host a biodiversity sometimes neglected in terms of knowledge if compared to that of caves (but see Fiasca et al., 2014; Mori et al., 2015; Di Lorenzo et al., 2018; Brancelj et al., 2020).

A notable contribution in assessing hotspots of species richness in caves and natural wells as means to entering proper caves has been for the first time presented by Culver & Sket (2000). The same authors used an arbitrary cutoff of 20 species for considering a cave a biodiversity hotspot by including both obligate terrestrial and aquatic subterranean species. In their approach they recognized the poor information available in the United States on the obligate cave-dweller aquatic microcrustaceans.

The availability of consistent numbers of records for non-cave subterranean habitats is scant. One exception is the large-scale sampling of karst and porous aquifers in Europe conducted as part of the PASCALIS project (Protocol for the ASsessment and Conservation of Aquatic Life In the Subsurface) (Gibert & Culver, 2009). The second notable case is the project DarCo (The vertical dimension of conservation: A cost-effective plan to incorporate subterranean ecosystems in post-2020 biodiversity and climate change agendas) which is intended to assess and manage the subterranean biodiversity across Europe under the climate change scenario (https://www.biodiversa.eu/2023/04/19/darco/). Several other studies put together cave and non-cave groundwater biodiversity (Dole-Olivier et al., 2009a; Stoch & Galassi, 2010; Iannella et al., 2020a; Iannella et al., 2020b; Iannella et al., 2021; Deharveng et al., 2024). Since the publication of the first hotspot list in 2000, knowledge of the global cave fauna has grown exponentially. For this reason, the original cutoff of 20 species was raised to 25 species by Culver et al. (2021), after cumulating terrestrial and aquatic species. Species lists are available for several tropical countries (e.g., Deharveng et al., 2021; Ferreira, Berbert-Born & Souza-Silva, 2023; Deharveng et al., 2023) and several caves throughout the tropics and sub-tropics are now well sampled (e.g., Li et al., 2022; Moutaouakil et al., 2024).

Unfortunately, knowledge about biodiversity in karst areas remains unbalanced, both geographically (i.e., more sampled areas versus less sampled areas) and in terms of the species collected (only terrestrial versus only aquatic, vertebrates versus invertebrates). This situation complicates the comparison of distinct biodiversity patterns, which depend on the different taxonomic groups analysed, and further reinforces the concept underlying the “Racovitzan Impediment” (Ficetola, Canedoli & Stoch, 2019) (i.e., the difficulty in exploring subterranean habitats, leading to incomplete knowledge of their biodiversity).

All of this makes difficult, if not impossible or misleading, the comparison among many groundwater habitat types in karst landscapes, managing different taxa in different habitats and geographic areas.

Another critical issue is the lack of long-term datasets and the scarcity of systematic monitoring programs. Historically, speleobiological research has been marginalized compared to other fields of the natural sciences, leading to delays in the development of theoretical and practical frameworks for the management and conservation of subterranean biodiversity. This situation hampers not only a comprehensive understanding of ongoing ecological and evolutionary processes, but also the formulation of effective conservation strategies based on solid and up-to-date knowledge.

The inherently hidden and fragmented nature of karst subterranean habitats poses numerous challenges for scientific research. Firstly, physical access to these environments requires advanced speleological techniques and limits the ability to conduct systematic and large-scale investigations. Consequently, knowledge of subterranean biodiversity remains fragmented and incomplete, with entire taxonomic groups awaiting to be discovered or formally described (Mammola et al., 2019a). Furthermore, the extreme fragmentation of habitats and their isolation foster speciation processes and high levels of endemism, complicating the interpretation of evolutionary relationships and biogeographic patterns.

From a conservation standpoint, groundwater biodiversity is increasingly exposed to intense anthropogenic pressures. Groundwater pollution resulting from intensive agricultural practices, industrial discharges, and urban infiltration severely compromises water quality and disrupts the ecological balance of groundwater habitats in karst and non-karst areas. Overexploitation of groundwater resources for agricultural, industrial, or civil uses can lead to piezometric declines and the drying of subterranean aquatic habitats, threatening the survival of GW taxa (Di Lorenzo & Galassi, 2013; Mermillod-Blondin et al., 2023; Fišer et al., 2025). On the other hand, these communities provide key ecosystem services, acting as important contributors to organic matter processing and the nutrient cycling through their ecosystem engineering activities (Boulton et al., 2008; Griebler & Lueders, 2009; Griebler & Avramov, 2015; Mammola et al., 2025) and the primacy is taken by the invertebrates stably living in groundwater (Mermillod-Blondin et al., 2023), as the small-sized copepods. Moreover, the ongoing effects of global climate change, although still poorly understood in these contexts, are likely to profoundly modify karst aquifer recharge patterns and alter the environmental stability essential for subterranean life (Mammola et al., 2019b; Sánchez-Fernández et al., 2021; Cerasoli et al., 2023; Vaccarelli et al., 2023b; Saccò et al., 2024).

In this study, we investigate, at the European scale, how much biodiversity in karst areas is represented in the cave subset and how much in other karst groundwater habitat types, with the aim of identifying karst biodiversity hotspots. We acknowledge, however, that in many karst areas the sampling effort is not consistent across different groundwater karst habitats, nor among caves within and between hydrogeological karst units. Despite these limitations in the current state of knowledge, we took advantage of the opportunity to analyse the biodiversity of karst groundwaters using a target group of microcrustaceans—the Copepoda—for which a recently published expert-curated database is available (Cerasoli et al., 2025). Copepods represent the most diverse group of crustaceans in groundwaters (Galassi, Huys & Reid, 2009) and are distributed across all types of groundwater habitats and microhabitats. Despite their wide distribution, they are considered part of the “unseen metazoans” due to their reduced body size (Galassi, Huys & Reid, 2009; Malard, 2022). We therefore adopted a “one taxon” approach, focussing on Copepoda, the group for which we have the largest number of records at European scale and sufficient knowledge of ecological specialization to groundwater, allowing us to discriminate the obligate groundwater dwellers from the occasional inhabitants of groundwaters (Galassi, Huys & Reid, 2009; Iannella et al., 2020a; Iannella et al., 2020b).

We hypothesize that cave habitats can serve as reliable surrogates for assessing the overall biodiversity of GW copepods in European karst regions; however, their representativeness may vary depending on regional factors such as cave density, sampling effort, and habitat heterogeneity.

Our study aims to: (i) assess whether karst biodiversity hotspots are better described by the caves located in the corresponding areas - by clumping copepod species richness of both the epikarst and the saturated karst for each cave, if present - or, alternatively, by karst habitats other than caves; (ii) determine whether and to what extent, within karst hotspots, cave species richness reflects the overall richness of the broader karst area, in order to assess whether cave biodiversity alone can adequately describe biodiversity hotspots in European karst regions.

Materials & Methods

Selection of occurrence records of groundwater copepods

The study area covers the European continent, main islands included (longitude min = −31.3°W, longitude max = 65.2°W; latitude min = 27.6°N, latitude max = 69.2°N). The area is a mosaic of 61,275 groundwater habitat patches, each representing one out of the three groundwater habitat types mapped by Cornu, Eme & Malard (2013) based on groundwater flow type, namely: aquifers in consolidated rocks, aquifers in unconsolidated sediments, and practically non-aquiferous rocks.

Only patches classed as aquifers in consolidated rocks were selected in the present study. This groundwater habitat type includes cave waters (both the unsaturated and the saturated karst in caves), karst springs (both intermittent and permanent over time), karst rivers and wells drilled in consolidated rocks across Europe (Fig. 1).

Figure 1 Examples of European karst GW habitats.

(A, B) GW-fed stream (Mostnica and Rak stream, Slovenia). (C) Basal spring (Presciano, Peninsular Italy). (D) Dripping pools in cave (Županova, Slovenia). (E) Subterranean lake (Tana che Urla cave, Peninsular Italy). (F) Subterranean stream (Pekel pri Zalogu, Slovenia). Photo credit: (A–D), (F) Emma Galmarini; (E) Andrea Massagli.

The occurrence records of GW copepods in each karst patch were retrieved from the EGCop dataset (https://zenodo.org/records/14608863), the first expert-validated, Europe-wide occurrence dataset for obligate groundwater-dwelling copepods (Cerasoli et al., 2025). The EGCop dataset comprises 6,986 records from 588 copepod species/subspecies distributed among four orders: Cyclopoida (3,664 records, 184 species), Harpacticoida (3,288 records, 395 species), Calanoida (32 records, seven species), and Gelyelloida (two records, two species). To perform geospatial analyses, we filtered the dataset by: (i) selecting only the records with spatial uncertainty in the associated coordinates lower than 10 km, this threshold being the one that best balanced the need for retaining sufficient occurrence data with that of excluding occurrences with too high spatial uncertainty; (ii) including only those records located within five km of the boundaries of polygons representing European aquifers in consolidated rocks, thus taking into account the spatial uncertainty both in the occurence records and in the mapped borders between aquifer in consolidated rocks and adjacent aquifer types. The records with uncertain taxonomic definition, indicated as “sp.”, were excluded from the analyses as it was not possible to evaluate whether within a given genus they corresponded to the same or to distinct species.

The resulting “karst” dataset was then further filtered to obtain a second dataset (“caves-within-karst”, hereinafter simply “caves”) containing only those karst records reported from caves.

Hot-spot analysis

To compute statistically significant groundwater biodiversity hotspots (Iannella et al., 2020a; Iannella et al., 2020b; Iannella et al., 2021) driven by species occurrences in karst and cave habitats, we first spatially joined each of the above-described datasets (“karst” and “caves”) to the vector data of groundwater habitat types from Cornu, Eme & Malard (2013). Each occurrence from each dataset was assigned to a habitat patch accordingly, and karst- or cave-related species richness was computed for each patch.

Then, we applied the Getis-Ord Gi* statistics (Getis & Ord, 1992) as implemented in ArcGIS Pro 3.4.3 (ESRI Inc., 2025) to identify hotspots (or coldspots) of species richness from karst and caves records. This method evaluates whether the spatial clustering of a biodiversity indicator (in this case, species richness for each patch) deviates significantly from a random distribution. The Gi* algorithm calculates a z-score and an associated p-value for each patch, based on the values of neighbouring patches and their distance (using Euclidean distance and an inverse distance weighting function). We classify patches with high z-scores and p-values above 95% confidence intervals as statistically significant hotspots.

The obtained z-scores were classed using the Jenks natural breaks optimization method (Jenks, 1967), which minimizes intra-class variance while maximizing inter-class variance. In line with Iannella et al. (2021), we focussed our interpretation on the “hottest hotspots”, defined as the top class of z-scores, corresponding to p-values ≤ 0.05. These patches reflect a significant non-random aggregation of high species richness which may serve as a prioritization proxy for conservation planning (Fleishman et al., 2006).

To quantify how much area within each karst hotspot is described by caves we calculated the percent coverage by cave hotspot patches within their karst counterparts (cave–within–karst coverage). All spatial analyses were conducted in ArcGIS Pro 3.4.3.

Results

After the application of a 10 km-filter of spatial uncertainty to the EGCop records associated with karst habitat types, 2,526 occurrence records were retained, representing 369 GW copepod species/subspecies (about 62% of the total number of species/subspecies currently included in the EGCop database; Table S1). Among these species, 228 belong to Harpacticoida (1,199 occurrences), 132 to Cyclopoida (1,293 occurrences), seven to Calanoida (32 occurrences), and two to Gelyelloida (two occurrences). A total of 131 out of 369 species are also present in aquifer types other than karst, and they could therefore be defined as obligate groundwater-dwelling generalists. On the other hand, 238 species (995 occurrences) are exclusively recorded within karst areas and have never been found in other aquifer types (karst specialists).

Records collected from caves are 1,867 (73.95% of the total number of karst records) and belong to 318 species (Harpacticoida = 189 species, 808 records; Cyclopoida = 122 species, 1029 records; Calanoida = seven species, 31 records), representing 86.1% of the total species richness of karst habitats (Table S1). Notably, 164 species are exclusive to caves (Table S1) meaning that these species were never recorded from other European karst habitats (e.g., karst springs, surface karst streams, and wells drilled in aquifers in consolidated rocks) or other groundwater habitat types (i.e., aquifers in unconsolidated sediments or practically non-aquiferous rocks).

The HSA of GW copepod species richness across European karst habitat patches revealed seven “hottest” hotspots defined by the highest p-value ranges (p-value ≤ 0.05 and p-value ≤ 0.01; Figs. 2A and 2C; Table 1). When considering only GW copepod species recorded from cave habitats within karst patches, seven statistically significant hotspots of GW copepod species richness were identified across Europe, partially overlapped to the previously detected karst hotspots but showing a slightly different spatial extension (Figs. 2B and 2D; Table 1).

Figure 2 Results of Getis-Ord Gi* statistics (HSA).

Gi* z-scores: (A) European karst areas, and (B) caves within karst areas (natural-break classes). Statistically significant (orange patches: p-value ≤ 0.05; red patches: p-value ≤ 0.01) hotspots of species richness: (C) karst areas, and (D) caves within karst areas. Seven main hotspot areas are detected from western to eastern Europe for both karst areas and caves: (1) the Cantabria Mountains, the Pyrenees and Hérault Massif; (2) the Jura Massif and south-western German uplands; (3) the central–southern Apennines; (4) Sardinia Island; (5) the eastern Alps together with the Lessinian Prealps, the Slovenian Dinarides and the External Dinarides extending from Croatia to Albania; (6) the Carpathians, and (7) the Balkan Mountains.

Table 1 Total extension of each hotspot macroarea, the spatial overlap between karst hotspots and cave-derived hotspots, and the percentage of cave hotspot coverage within the respective karst hotspot area.

Hotspot macroarea	Area–Karst
(km2)	Area–Cave
(km2)	Cave hotspot
area within karst
hotspot	Percentage of cave
hotspot area within
karst hotspot	
Cantabria–Pyrenees Mountains–Hérault Massif
(Spain, France)	45,952.977	45,299.349	43,043.16	93.73%	
Jura Massif
(France)	33,137.313	22,543.574	22,360.153	67.48%	
Island Sardinia
(Italy)	259.274	259.274	259.274	100%	
Central Apennines
(peninsular Italy)	12,455.544	4,574.677	4,574.677	36.73%	
Eastern Alps–Lessinian
Prealps–Dinarides
(Italy, Slovenia, Croatia to
Albania)	68,819.635	65,487.251	65,190.859	94.73%	
Carpathians Mountains
(Slovakia, Romania)	99,978.230	56,508.573	56,508.573	56.52%	
Balkans Mountains
(Serbia, Bulgaria)	54,130.760	52,783.354	51,690.844	95.49%	

The smallest hotspot, corresponding to Sardinia Island, showed the largest overlap (100%) between the karst-based hotspot polygons and the cave-based ones, while for the largest hotspot, the Carpathians Mountains, the overlap between karst and cave hotspots dropped to 56%. The lowest percent overlap between the two hotspot types was found in Central Apennines, amounting to 36% (Table 1). Harpacticoida represented the most contributing order, in terms of species richness, within four karst-based hotspots (Cantabria and Pyrenees Mountains, Hérault Massif; Central Apennines; Eastern Alps-Lessinian Prealps-Dinarides; Balkan Mountains), while Cyclopoida were slightly preponderant in the Jura Massif (Table 2); in the remaining hotspots, the contribution of the two orders was almost equal. Similar patterns emerged for cave-based hotspots except for Sardinia, where Harpacticoida showed a tripled contribution (six species) compared to Cyclopoida (two species) (Table 2).

Table 2 The total number of species defining the karst-based and cave-based hotspots and distribution across the four copepod orders.

The last column reports the percentage ratio between cave-hotspot species and karst-hotspot species.

Hotspot macroarea	Karst hotspot species	Cave hotspot species	% Cave species/
Karst species	
	Total	Calanoida	Cyclopoida	Gelyelloida	Harpacticoida	Total	Calanoida	Cyclopoida	Gelyelloida	Harpacticoida		
Cantabria–Pyrenees Mountains–Hérault Massif
(Spain, France)	88	1	35	1	51	75	1	33	0	41	85.22%	
Jura Massif
(France)	29	0	15	1	13	23	0	13	0	10	79.31%	
Island Sardinia
(Italy)	10	0	5	0	5	9	1	2	0	6	90%	
Central Apennines (peninsular Italy)	24	1	4	0	19	9	0	5	0	4	37.50%	
Eastern Alps–Lessinian Prealps–
Dinarides (Italy, Slovenia,
Croatia to Albania)	114	3	41	0	70	94	3	32	0	59	83.10%	
Carpathians Mountains
(Slovakia, Romania)	30	0	15	0	15	29	0	15	0	14	96.66%	
Balkans Mountains
(Serbia, Bulgaria)	54	0	21	0	33	46	0	22	0	24	84.90%	

Table S2 provides the distribution patterns of the species that characterize the identified hotspots.

Karst hotspots in Europe

The karst hotspot located in western Europe embraces most of the Cantabria Mountains, the Pyrenees, reaching the Hérault Massif, just mirroring the stygodistrict I3 (Pyrenean-Aquitanian Province according to Botosaneanu, 1986). This hotspot macroarea is defined by 88 species, with Antrocamptus catharinae, A. chappuisi, and A. coiffaiti marking the Pyrenean area. The monotypic cyclopoid genus Kieferella, with the species K. delamarei, Graeteriella vandeli, the ectinosomatid harpacticoid Pseudectinosoma vandeli, the extraordinary gelyelloid Gelyella droguei are asssociated to the portion of this hotspot corresponding to the Cent-Fonts karst system along the Hérault river, with Stygepactophanes occitanus which marks the easternmost part of this karst macroarea.

The central European hotspot macroarea is represented by the Jura Massif and surrounding areas. The widespread GW harpacticoids Ceuthonectes gallicus and C. serbicus, the parastenocaridid Fontinalicaris fontinalis fontinalis and the cyclopoid Monchenkocyclops biarticulatus, which is also known from the hyporheic zone of streams in Portugal and from springs and caves in Spain, are all associated to this hotspot. Furthermore, several representatives of the genus Speocyclops define this hotspot, together with the gelyelloid Gelyella monardi collected from a karst spring in the Swiss Jura and the canthocamptid harpacticoid Stygepactophanes jurassicus, the latter two being the most exclusive species of this hotspot.

The central Apennines in the Italian Peninsula along with a few satellite areas in southern Italy, represent the southernmost hotspot macroarea in Europe. It encompasses 24 GW copepods and is mainly characterized by the presence of Acanthocyclops agamus known only from two karst springs and one syphon lake within the hotspot (Galassi & De Laurentiis, 2004; Di Lorenzo et al., 2018). The widely distributed Diacyclops cosanus is another marker of this macroarea, being collected from several groundwater habitat types including high salinity alluvial aquifers, true fresh groundwaters, and saturated sulfidic karst of the Frasassi Cave (Galassi et al., 2017). Among harpacticoids, worth of mention are the phyllognathopodid Phyllognathopus inexspectatus, the only GW species of the genus (Galassi, De Laurentiis & Fiasca, 2011), the ameirids Nitocrella pescei and N. kunzi with the parastenocaridid Simplicaris lethaea, all being representative species of this hotspot. The ameirid Parapseudoleptomesochra italica is widely distributed in the Italian Peninsula and known also from a well in Switzerland (Moeschler & Rouch, 1984) and from the saturated karst of the Movile Cave (Romania) (Brad, Iepure & Sarbu, 2021). The ectinosomatid Pseudectinosoma reductum is distributed in karst habitats of this hotspot and known from the sulfidic karst of the Melissotrypa Cave (central Greece) (Popa et al., 2019). Among the Calanoida, a GW population of Eudiaptomus cf. intermedius has been discovered in the Frasassi Cave. This population has a controversial position and its attribution to the surface species E. intermedius is still open to question (Galassi et al., 2017).

The island of Sardinia has a small hotspot area in its central-eastern part, close to the Tyrrhenian shoreline, which is mainly defined by the GW copepods collected in the Bue Marino Cave. The copepod diversity of this area is only partially known, and the copepods recorded so far are predominantly represented by freshwater species, such as the ameirid harpacticoid Nitocrella beatricis which has been found also in Corse, in the hyporheic zone of rivers and streams in Nuoro and Cagliari provinces, and in wells within the small islands of Tavolara, Molara, Caprera, and La Maddalena (Cottarelli & Bruno, 1993; Cottarelli, Bruno & Forniz, 1996). Among the other species associated to this hotspot, worth to mention are the canthocamptid Ceuthonectes pescei, Elaphoidella janas, the parastenocaridid Parastenocaris triphyda, an undescribed species of the genus Schizopera, the cyclopoid Speocyclops sardus and Metacyclops trisetosus. The latter has a disjunct distribution, being recorded from the Bue Marino Cave, the “core” of the Sardinian hotspot, but also from Antro di Bagnoli (Italian Eastern Alps) and from the Aesculapius Cave (Croatia). Moreover, a karst hotspot area (95% confidence) is present in NW Sardinia which corresponds to karst landscapes of Capo Caccia-Punta Giglio, and is related to some copepod species collected from Dasterru di Punta Giglio (Alghero) and the Nettuno Cave.

The hotspot with the greatest species richness embraces a large portion of the Italian Eastern Alps and the Slovenian Dinarides (Table 2) that together define the “Classical Karst” (Ford, 2004; Jurkovšek et al., 2016). This hotspot macroarea includes also the Lessinian Prealps in Italy and the external Dinarides southward to Albania. In the present study, 114 GW copepods are associated with this hotspot macroarea. Some GW species of the cyclopoid genus Acanthocyclops occur both in caves and in other karst habitats of this hotspot (e.g., A. troglophilus, A. hypogeus), whereas A. gordani and A. kieferi are recorded also from other groundwater habitat types (e.g., hyporheic zones). Diacyclops charon, D. slovenicus and D. tantalus are exclusive to this karst hotspot whereas many other species of the genus Diacyclops are found in this hotspot and in several groundwater habitat types of Europe. Speocyclops infernus marks this area despite it is also found in the Balkan karst in Bulgaria. Among harpacticoids, the canthocamptid Ceuthonectes pertkovskii and C. rouchi are linked to the Slovenian karst, whereas C. serbicus is widespread in this hotspot and in other karst and non-karst regions of many European countries (France, Italy, Switzerland, Slovenia, Romania, Bulgaria, Serbia, Macedonia, Hungary, and Georgia). Twenty GW species of Elaphoidella have been recorded in this area, representing approximately 30% of the species richness described within the order Harpacticoida. The genus Lessinocamptus, with three described species, defines the subarea of the Lessinian Prealps (Vaccarelli et al., 2023a) extending to the eastern Italian Alps and reaching the Slovenian karst with one record of an undescribed species, Lessinocamptus sp. SLO1. One of the three species described so far in the genus Spelaeocamptus, S. incertus, is represented in this hotspot area. The genus Morariopsis with three species and the monotypic genus Paramorariopsis contribute to defining this hotspot, as they are found only in this hotspot and have never been found in non-karst areas. The diversification observed in the ameirids (three species of Nitocrella and P. italica) is lower than in the canthocamptids. The Parastenocarididae are represented by two genera (Horstkurtcaris and Parastenocaris), reaching a total of 13 species/subspecies. The GW Calanoida which are represented by seven species in European groundwaters, have three representatives in this area, belonging to two genera: Stygodiaptomus (two species) and Troglodiaptomus (one species). They are always linked to karst aquifers and never found in other groundwater habitat types.

The Carpathian Ridge is the largest hotspot detected by the HSA, and it is defined by 30 GW copepod species. Among the cyclopoids, nine species of Acanthocyclops are included in this hotspot, followed by four Diacyclops species, Graeteriella unisetigera—which is widespread throughout Europe and also occurs as a cryptozoic element (Fiers & Ghenne, 2000)—and Speocyclops troglodytes. The latter shows a disjunct distribution encompassing Italy, Romania and Serbia, and is predominantly found in caves but also in hyporheic habitats in northeastern Italy. Among harpacticoids, species richness is dominated by the Canthocamptidae family, wich includes four species of the genus Elaphoidella as well as Ceuthonectes hungaricus—which defines this area with five occurence records—and C. serbicus, notable for its broad distribution across Europe. The Parastenocarididae family is also well represented, with four species of Parastenocaris and Stammericaris phreatica. Similarly, Speleocamptus spelaeus has 29 records in Romania, primarily linked to karst habitats, although two records come from alluvial aquifers within the same country.

The hotspot covering the Balkan Mountains is defined by 54 GW copepod species. Among cyclopoids, the genus Acanthocyclops has the primacy with nine species (of which A. balcanicus balcanicus, A. chappuisi, A. iskrecensis, A. radevi, A. reductus, A. strimonis are exclusive to this macroarea), followed by Speocyclops with six species (of which S. lindbergi, S. proserpinae and S. plutonis are only known from this hotspot) and Diacyclops with four species (of which D. haemusi and D. fontinalis are known only from this hotspot). Among harpacticoids, Elaphoidella is the most representative genus, with 15 species recorded in this hotspot. Worth mentioning is the presence of the canthocamptid Ceuthonectes haemusi which marks this area, followed by the widespread C. serbicus. The harpacticoid ameirids are represented by two species of Nitocrella, two species of Parapseudoleptomesochra, and Nitocrellopsis intermedia which is linked to this area. The Parastenocarididae are represented by five species (of which Parastenocaris curvicaudata, P. jeanneli, P. karamani karamani mark this hotspot).

Cave hotspots in Europe

The cave hotspot embracing the Cantabria Mountains, the Pyrenees and the Hérault Massif overlaps with the karst hotspot by 93.73% (Table 1). This hotspot is defined by 75 species, including some species known only from cave habitats, such as the harpacticoid Cottarellicaris gallicus, Parapseudoleptomesochra subterranea deminuta, Proserpinicaris cantabrica. Among the cyclopoids, unique cave species are Speocyclops arregladensis and S. racovitzai guillounensis.

The cave hotspot of the Jura Massif overlaps with the corresponding karst hotspot by 67.48% (Table 1) and mostly overlaps with the southern part of the corresponding karst hotspot (Fig. 1D). This hotspot encompasses 23 species. Caves in this area host several generalist GW copepods among cyclopoids and harpacticoids, with a few exceptions for some endemics and not yet described species of the harpacticoid Bryocamptus and the cyclopoid Speocyclops.

The cave hotspot of the central Apennines shows the lowest overlap with its corresponding karst hotspot (36.73%, Table 1), likely due to the poor availability of cave records in the Abruzzi and Latium regions (Figs. 1C, 1D) which therefore did not emerge as a portion of the cave hotspot. This cave hotspot is defined by nine species; among them, the stygomorphic population of Eudiaptomus cf. intermedius stands out. Apart from Stammericaris lorenzae, all the species included in this hotspot are recorded from the Frasassi Cave (Galassi et al., 2017), meaning that the hotspot is almost exclusively defined by this large cave system.

The small cave hotspot in the Island of Sardinia coincides with the karst hotspot, showing a 100% overlap. This overlap is due to the presence of several copepod species found in the Bue Marino cave, including the canthocamptid Ceuthonectes pescei, Elaphoidella janas, the parastenocaridid Parastenocaris triphyda, and the cyclopoids Speocyclops sardus and Metacyclops trisetosus.

The hotspot macroarea identified as the Eastern Alps and Dinarides represents the largest cave hotspot detected in our analysis, representing 94.73% of the total area of the karst hotspot. This large cave hotspot is mainly defined by: a mix of cyclopoid species belonging to the cyclopoid Acanthocyclops and Speocyclops exclusive to caves; the harpacticoid species of Lessinocamptus and Moraria, Morariopsis kieferi and M. scotenophila, Paramorariopsis anae, Ceuthonectes species (with C. rouchi associated to the Slovenian karst, caves included), 17 GW species of the canthocamptid Elaphoidella, and 11 parastenocaridids, belonging to three genera (Parastenocaris, Horstkurtcaris, Italicocaris). Furthermore, this cave hotspot hosts three GW diaptomid calanoids (Stygodiaptomus kieferi, S. petkovskii, and Troglodiaptomus sketi), as free swimmers in the planktonic habitats of the subterranean lakes of the saturated karst.

The hotspot macroarea defined by the Carpathians caves covers 56.52% of the karst hotspot. The cave hotspot is described by 29 species, which mirrors to some extent the taxonomic diversity of the overall karst hotspot, with the primacy taken by the cyclopoid genus Acanthocyclops which includes several species found exclusively in caves. Members of Diacyclops are present both in caves and in other karst habitats; some other species are widespread across different groundwater habitat types encompassing the hyporheic zone and the alluvial aquifers (e.g., Diacyclops belgicus). A few species are not exclusive to caves and occurr in other karst habitats, such as Speleocamptus spelaeus which dwells in several caves but was also recorded in a groundwater-fed spring and in an alluvial aquifer in Romania. Conversely, Speleocamptus incertus is known with only one record from a cave in Macedonia (into the cave hotspot of the Eastern Alps-External Dinarides).

In the Balkan macroarea, the cave hotspot covers 95.49% of the karst hotspot. It is described by 46 species, mostly the same ones defining the karst hotspot. This suggests that, in the Balkan region, most GW copepods diversity is found in caves. Nine GW species of the cyclopoid genus Acanthocyclops are known from the karst hotspot, with only A. milotai being exclusive to the cave habitat. Among harpacticoids, Bryocamptus borus is only known from caves from two different karst areas (Slovenia and Serbia), Elaphoidella stygia is known from two caves only in Bulgaria, and Ceuthonectes haemusi is recorded from two caves within this hotspot.

Discussion

The subterranean biosphere is increasingly recognized as a global biodiversity frontier. Karst aquifers and cave systems, with their structural complexity, high endemism, and functional specialization, are emerging as biodiversity hotspots rather than barren voids. Recent discoveries—both in remote and seemingly well-known areas—continue to increase cave species richness, reinforcing the idea that a single cave can represent a localized biodiversity hotspot (Souza-Silva & Ferreira, 2016; Pipan, Deharveng & Culver, 2020; Culver et al., 2021; Huang et al., 2021; Niemiller, Helf & Toomey, 2021; Mammola et al., 2022; Gallão et al., 2023; Hernández-Lozano et al., 2024).

Despite increasing attention to biodiversity in global policy, groundwater ecosystems remain poorly integrated into conservation frameworks (Iannella et al., 2020a; Iannella et al., 2020b; Iannella et al., 2021; Sánchez-Fernández et al., 2021; Wynne et al., 2021; Vaccarelli et al., 2023b; Mammola et al., 2024; Rohde et al., 2024; Saccò et al., 2024). In Europe, karst regions like the Dinaric Karst, the Alpine arc, Carpathians, Balkans, Iberian and Apennine massifs host unique obligate groundwater-dwelling taxa shaped by long-term isolation. Comparable diversity patterns occur globally—in the Anatolian Plateau, Appalachians, Southeast Asia, and Neotropics—forming a mosaic of subterranean aquatic hotspots (Christman et al., 2016; Deharveng et al., 2021; Souza-Silva et al., 2021).

However, much of this biodiversity remains undocumented. Sampling subterranean meiofauna, particularly GW copepods, is hindered by microscopic size, taxonomic impediments, and low visibility (Giere, 2008; Ficetola, Canedoli & Stoch, 2019). Cryptic diversity and under-described taxa (Bron et al., 2011; Karanovic, Djurakic & Eberhard, 2016) contribute to significant knowledge gaps and extinction risks, leading to the so-called “Centinela extinctions” (Wilson, 1999) (i.e., the disappearance of a species before it could be described). Indeed, copepod species richness is projected to increase markedly by 2100, with over 90% of freshwater copepods being endemic to single zoogeographic regions (Macêdo et al., 2024).

Sampling bias also affects regions like Mediterranean islands, southern Italy, and Greece. Unpublished data and unexplored karst habitats suggest underestimated diversity, as exemplified by Galmarini et al. (2023), who found numerous undescribed copepod species across southern Italian caves.

The microscopic and hidden nature of many groundwater taxa limits both scientific investigation and conservation attention, despite their vital roles in ecosystem functions like nutrient cycling and water purification (Boulton et al., 2008; Mermillod-Blondin, 2011; Griebler & Avramov, 2015; Howard et al., 2023). Taxonomic bottlenecks (Culver & Sket, 2000) and low public appeal relative to invertebrates (Hutchins, 2018; Oliveira & Ferreira, 2024) further marginalize them. Nonetheless, karst meiofauna include unique evolutionary relics of high conservation value (Galassi, Huys & Reid, 2009; Fattorini et al., 2020; Sánchez-Fernández et al., 2021).

In this contribution, we analyzed copepod diversity patterns within European karst areas and associated cave systems, to identify spatial conservation priorities. Our findings support the view that karst groundwater ecosystems are biodiversity cores, essential to inclusive and effective environmental governance (Saccò et al., 2024).

The dialogues between karst areas and caves

GW copepods are recorded from any groundwater habitat type across Europe (Cerasoli et al., 2025). In karst groundwaters, they occur in all habitats and microhabitats.

Previous studies, analysing occurrence records of GW harpacticoid copepods across all the groundwater habitat types of Europe (Iannella et al., 2020a; Iannella et al., 2020b; Iannella et al., 2021), demonstrated the suitability of these organisms in delimiting European groundwater biodiversity hotspots. The results obtained in the abovementioned studies are surprisingly convergent, to some extent, to the ones emerging in the present analyses. For instance, the species richness hotspots in the Pyrenees, in the Eastern Italian Alps and Dinarides, and in Central Apennines that were found in Iannella et al. (2020a) are included in the “karst” and “caves-within-karst” hotspots we define here.

The total number of GW copepods species occurring in karst areas of Europe amounts to 369. Among these latter, 238 species have been found exclusively in karst habitats (caves included) and can thus be defined as karst specialists. However, this figure underestimates the real number of European karst specialists, pending the description of several new species. Moreover, other karstic species may have been excluded after the spatial filtering adopted for the HSA, due to the geographical uncertainty of some records and/or the absence of details for the locality data. A subset of these karst specialists, 164 species (∼68%), have been collected exclusively in caves and can thus be defined as cave specialists. The remaining 74 karst specialist species have instead been recorded from karst habitats other than caves. Consequently, the other groundwater karst habitats in the karst regions play an important role in explaining the total species richness of the karst. Finally, the 131 species occurring both in karst and in non-karst areas can be defined karst generalists, as they also occur in the hyporheic zone of streams and rivers, in alluvial aquifers, in practically non-aquiferous rocks, alluvial springs, in the upwelling zone of lakes or in the hypothelminorheic habitats (Meštrov, 1962; Culver, Pipan & Gottstein, 2006). In this case, the boundaries between different aquifer types work as transmissive borders for GW copepods (Iannella et al., 2020b).

The hotspot analysis performed considering only the copepod species occurring inside the caves within karst areas highlights the same number of hotspots (seven) as when extending the analysis to all karst records. Furthermore, these cave-defined hotspots fall within the geographical limits defined by the karst ones.

The overlap of cave hotspots with the karst ones is different among the detected hotspots macroareas. Several factors may contribute to explaining this pattern. First, not all the karst regions have the same number of caves per unit area; second, usually not all the caves located in a certain region are known, as some of them await to be discovered; third, not all the known caves have been sampled; fourth, not all the caves have the same mesohabitat heterogeneity, such as the divide between the epikarst/unsaturated karst, with its characteristic mesohabitats (gours, dripping pools, temporary siphons, trickles) (Pipan et al., 2018), and the saturated karst, which may have contrasting tridimensional morphologies (vertical shallow or deep wells, large and small lakes more or less interconnected, perennial cave streams and springs). With respect to the latter point, younger caves tend to exhibit less developed, and therefore less heterogeneous, mesohabitats due to the relatively recent karstification process. Indeed, in younger caves, the chemical and physical processes that create different mesohabitats have had less time to operate, resulting in a more uniform environment. As cave development and karstification progress over time, the physical and chemical processes generate increasingly diverse features (such as breakdown, vadose zones, and different water flows), increasing the complexity and heterogeneity of the mesohabitats within a cave (Moldovan, Kováč & Halse, 2018; Balogh et al., 2020; Cardoso, Ferreira & Souza-Silva, 2022; Petrovová et al., 2024).

The cave legacy

Among the karst hotspots identified in our study, some are clearly defined by the copepod species richness known from caves, which works as a good proxy for the overall high species richness found in the corresponding karst hotspots. This condition has been found in: (1) the western hotspot macroarea represented by the Cantabria Mountains (with over 4,000 caves) together with the Spanish and French Pyrenees (known to host thousands of caves, precise information on the total number of caves being unavailable); (2) the central-eastern hotspot macroarea which embraces the Eastern Alps (whose easternmost sector bordering Slovenia hosts 8,677 caves, https://catastogrotte.regione.fvg.it/) extending southward to the Lessinian Massif (with 1,600 caves, Peresani & Sauro, 2024), the Slovenian Dinarides hosting 15,884 caves with a mean of 2 caves per km2 (https://www.katasterjam.si/; UNESCO World Heritage Convention, 2015), and the external Dinarides with about 25,000 caves (Zagmajster et al., 2010); (3) the Balkan karst hotspot where more than 4,500 caves are known (Cave Rescue Bulgaria, 2025).

Referring to the Eastern Alps-Lessinian Prealps-Dinarides cave hotspot, it is defined by 94 species, mainly found in the “Classical Karst”, reinforcing evidence from previous studies (Sket, 1999; Culver & Sket, 2000; Brancelj & Pipan, 2004; Culver et al., 2004; Pipan & Culver, 2007; Pipan et al., 2018; Brancelj et al., 2020; Zagmajster, Polak & Fišer, 2021; Galmarini et al., 2023). The “Classical Karst” is the most investigated European area in terms of subterranean biodiversity, being consistently claimed to represent a hotspot in terms of groundwater species richness and endemism (Sket, 1999; Culver & Sket, 2000; Brancelj et al., 2020; Iannella et al., 2020a; Deharveng et al., 2024). Indeed, the Dinaric caves, together with the ones of the Eastern Alps and the Lessinian Prealps are the “pulsating heart” of GW biodiversity and copepod species richness of Europe.

Somehow differently form the abovementioned areas, the Central European hotspot—represented by the Jura massif—does not have a well-defined density of caves. However, it is distinctive in that the known caves are often part of well-developed karst systems (Durlet et al., 2024). The Jura cave hotspot is relatively large if compared to the total hotspot karst area, indicating high cave habitat availability of recent origin in the area, which was also likely affected more intensively by the Riss-Würm effect (Castellarini et al., 2007; Dole-Olivier et al., 2009b).

Caves as islands

At a first glance, the concept of “island” applied at a local scale to individual caves (Mammola, 2018; Culver & Pipan, 2019; Balogh et al., 2020) has limits related to the hydrological connectivity that may exist between single caves being close to each other or belonging to the same hydrogeological unit. However, this condition does not always occur. In many cases, even when a hydrological continuum exists, species are not shared between adjacent caves, and perhaps more surprisingly, not even among different mesohabitats within a single cave. For example, in Eastern Alps-Lessinian Prealps-Dinarides hotspot, cave habitat availability (Christman & Culver, 2001) has the primacy in favoring colonization, isolation and speciation of copepod species, where sometimes any dripping pool hosts its own copepod assemblage. In such cases, any cave works as an archipelago, with pools, trickles, gours, dripping pools, micro-fractures in the vadose zone representing islands into the cave archipelago, or islands in the main island represented by one cave (Pipan et al., 2018; Balogh et al., 2020; Cardoso, Ferreira & Souza-Silva, 2022). Usually, the unsaturated portion of a cave does not share copepod species, except for a few generalist GW species, with the saturated portion of the aquifer in the same cave.

Certainly, the lack of long-term monitoring data for most subterranean environments can, to some extent, alter the interpretation of available observations. Nonetheless, evidence from well-known caves with a good sampling effort clearly shows that GW copepods, perhaps also due to the poor tendency to dispersal in some species (especially among the harpacticoids and some inbenthic cyclopoids) or to extreme specialization for specific mesohabitat (e.g., some planktonic calanoids), are often unable to establish permanent populations in other caves and fail to reproduce there. A trend which corroborates the view of caves as “biodiversity islands”, at least for groundwater meiofauna. Thus, the statement by Lamoreaux (2004) that GW microcrustaceans may easily disperse by means of passive dispersal remains debatable because many copepods may not (Galassi, Huys & Reid, 2009).

Hotspots in European karst areas: geological and climatic drivers

Most of the karst hotspots we highlighted fall close to the border of the Last Glacial Maximum (LGM, ∼21 kyBP; Becker et al., 2015), particularly the western European karst macroarea embracing the Hérault massif as eastern limit, the “Classical Karst” embracing the Eastern Alps and the Dinarides, the Central Apennines and the Sardinian hotspots, the Carpathian and Balkan ones in southeastern Europe. The last Quaternary glaciation certainly left residual effects on the distribution of species in subterranean environments. The largest clusters of underground species richness, for both terrestrial and aquatic taxa, are indeed found in areas of southern Europe that functioned as refuges (Galassi et al., 2009; Stoch & Galassi, 2010; Zagmajster et al., 2014; Iannella et al., 2020a; Vaccarelli et al., 2023a).

Europe’s GW copepod hotspots likely represent the diachronous yet interconnected expression of a peri-Tethyan carbonate “engine”, whose successive tectono-sedimentary “awakenings” may account for the tightly clustered distribution of these hotspots along a latitudinal belt between 40°N and 50°N. Specifically, Late Variscan hydrothermal dolomitisation first seeded a porous substrate in Sardinia (∼300–280 Ma) (Boni et al., 2000); then, Mesozoic platform aggradation along widening Tethyan margins created a continent-scale lattice of primary porosity subsequently reorganized by Jurassic–Cretaceous syn-rift faulting (Scheibner & Speijer, 2008), and Alpine–Carpathian convergence further fractured the carbonates, producing more pronounced hydraulic gradients. Finally, during the Middle–Late Miocene, back-arc slab rollback behind the Apennine–Carpathian fronts accelerated erosional unroofing (Haidar et al., 2022). Since most of these structurally rejuvenated aquifers lie south of 45°N—beyond the maximum extent of continental ice during the Last Glacial Maximum (Clark et al., 2009)—Quaternary glacio-eustatic oscillations contributed primarily to meltwater-driven segmentation rather than wholesale glacial sterilization. The resulting hydrogeological and climatic “sweet spot” aligns precisely with independent hotspot analyses of karst, cave and overlapping habitats that consistently pinpoint the 40–50°N belt (Deharveng et al., 2009; Stoch & Galassi, 2010; Zagmajster et al., 2014) and, at a finer scale, the 42–46°N ridge identified for cave terrestrial invertebrates (Culver et al., 2004) and for temperate-zone subterranean hotspots worldwide (Culver et al., 2021). Hence, the spatial concordance between geological activation pathways and biogeographical patterns suggests that Europe’s copepod-rich aquifers are the outcome of long-term carbonate evolution modulated by mid-latitude palaeoclimate dynamics.

Other palaeogegraphic and palaeoecological events left their mark in southeastern Europe, such as the fragmentation of the Paratethys in surface brackish basins and the subsequent disappearance of some of them. Gradual growth of the Alpine–Carpathian–Dinarides orogenic system during the Miocene induced progressive regression of the Western, Central and Eastern Paratethys. This geodynamically controlled paleogeographic and biogeographic differentiation is generally defined on the basis of characteristic faunal assemblages (mainly mollusks, foraminifers, and ostracods), which are mostly endemic to the Paratethys Sea. The Eastern Paratethys likely represented the primary center of differentiation for many aquatic species. Its disappearance, which began before and concluded during the Messinian salinity crisis (Lazarev et al., 2020), may have caused local extinctions of some taxa and facilitated colonization by others within subterranean environments. This pattern has been suggested for the aquatic diversity of the Movile Cave (Romania) (Brad, Iepure & Sarbu, 2021) located in the Carpathians and the northern bord of the Balkan Ridge.

The contribution of karst habitats versus caves in describing hotspots of copepod species richness

Karst areas are undoubtedly those that have best served the role as refuge for groundwater taxa, thanks to their fractured hydrogeological network that facilitated the vertical entry of surface-water ancestors, favouring also vicariance.

European karst hotspots vary in size, and cave density within these hotspots varies from area to area. The highest concentrations of caves in Europe are found in Cantabria, the Pyrenees, the Alps and Dinarides, the Balkans, and, to a lesser extent, in central and southern peninsular Italy.

The area covered by a certain karst system can offer a rough idea of how many species can be hosted therein, because this calculation, which is also normally used to identify protection zones for subterranean biodiversity, always deals with surface areas. What is still missing is the urgent assessment of the vertical dimension of the karst, with all its horizontal ramifications at various depths (Mammola et al., 2024) which may represent different mesohabitats, home to diverse invertebrate species.

We observed that when a karst hotspot shows a substantial overlap with its corresponding cave-based hotspot, the species recorded in caves are a good proxy of the overall GW species richness within that karst area, of whatever extension. For instance, in the Eastern Alps-Lessinian Prealps-Dinarides, 114 species define the karst hotspot, and 94 of them are also found in the corresponding cave hotspot, representing the 82.45% of the total species richness for that karst area. Similar patterns emerge for the Balkan and the Western European (Cantabria-Pyrenees-Hérault) hotspots. In all the aforementioned cases, caves are enough. The situation changes in Central Apennines, where the karst hotspot is poorly represented by its cave counterpart: caves account only 37.5% (nine species) of the total species recorded in this karst area (24 species). The explanation may be traced in the lower number of caves available in the area, and in the fact that several of the ones already known are still unexplored. In this case caves alone are not enough. Other karst habitats play a pivotal role as refugia for GW copepods, such as karst springs. Springs condense the conservation logic of subterranean ecosystems into small, manageable units: they are biodiversity hotspots forged by stable yet heterogeneous environmental regimes, refugia under increasing climate change, and practical gateways for monitoring aquifer health (Cartwright & Johnson, 2018). Protecting springs for the sake of obligate groundwater species demands a shift from site-by-site fixes to aquifer-aware networks that maintain hydrologic processes, leverage bioindicator sensitivity, and anticipate climate risks (Cartwright et al., 2020; Cerasoli et al., 2023). Their conservation value for subterranean aquatic biota (Fattorini et al., 2016) emerges from three intertwined properties: connectivity with underground networks, environmental stability coupled with microhabitat heterogeneity (Fiasca et al., 2014), and accessibility for monitoring and stewardship that is rarely achievable underground (Cantonati et al., 2020).

Apart from regions where speleogenesis did not lead to the formation of many caves, as in the central Apennine ridge, an area effect is apparent: where the overlap between karst and cave hotspots is extensive, caves tend to be representative of the whole karst GW copepod richness, such as in Sardina (100% overlap, one cave describes the whole karst hotspot). The island has a complex geological history (De Waele & Grafitti, 1998; De Waele & Grafitti, 2004), and many caves are distributed in spotted karst areas (covering about 8% of the island’s total surface) in central-east Sardinia and in the Cambrian Iglesiente Sulcis region in southwest. Other interesting karst landscapes are found in the northwest of the Island (Capo Caccia-Punta Giglio) and in the central eastern part (Tacchi area). This situation supports the contention that, at present, the copepod diversity of Sardinia’s karst groundwater is still poorly known. So, a strong sampling bias likely affects the extension of the hotspot identified through our geospatial analysis.

The Carpathians cave hotspot is an exception to this trend: despite the cave hotspot covers only about half the area of the corresponding karst hotspot (56.52% overlap), it harbours 96% of the species defining the karst hotspot. This suggests that caves in this area are richer in species than the other karst groundwater habitats: their species richness is additive, with many caves likely working as “islands”. In this unique case caves are much more than enough.

Implications for conservation

The EU Biodiversity Strategy for 2030 highlights the need for integrated, ecosystem-based approaches to protect biodiversity, water resources, and climate resilience. In this framework, it is recognized the dual role of karst aquifers as biodiversity reservoirs and vital water sources. Their conservation is aligned with key UN Sustainable Development Goals (SDGs 6, 13, and 15). The present contribution seeks to articulate a framework for the critical analysis of karst groundwater systems, positioning them at the core of global subterranean biodiversity. By highlighting their conservation value, we argue that karst groundwaters should not be viewed as peripheral anomalies, but rather as key components of biodiversity that are essential to inclusive and effective environmental governance.

By elucidating spatial patterns of GW copepod species richness, we delimit hotspots across European karst areas, then focusing on caves into such areas to highlight overlaps and mismatches between “karst” versus “caves-within-karst” hotspots. So far, the most effective approach to conserving biodiversity has focused on individual caves, based on the species richness they host. In many cases, this approach may work well when the karst area is predominantly characterized by cave development. In such cases, integrating hydrogeological knowledge about connections among neighbouring caves would be crucial for conservation. In short, it is better to conserve two hydrogeologically isolated caves rather than two connected ones, because in the latter case some species may disperse in both directions, thus making the two caves similar in terms of assemblage composition.

We identify seven main karst hotspots of GW copepod species richness across Europe. By replicating the hotspot analysis only on records collected from caves within the target karst areas, we derive that, in most cases, caves are good representatives of the species richness found in the overall karst hotspot unit. Thus, caves have a good potential to be targeted as priority sites for protecting groundwater biodiversity in karst regions, reducing the extent over which conservation measures should be addressed, although this should not be taken as a rule. The identification of GW biodiversity hotspots in karst areas across Europe certainly supports spatial planning and the identification of appropriate conservation measures. However, when considered in isolation this approach also has limitations. Groundwater is more or less closely connected to all surface water bodies, be they glaciers, snowfall, or rainfall, albeit to varying degrees. Therefore, protecting a single cave is not sufficient if anthropogenic pressures occur within the recharge surface of the cave’s aquifer, generating potential or actual impacts and ultimately contaminating the groundwater where GW species live. Indeed, the high specialization of many GW lineages to the groundwater environment, their extreme endemism, and the trophic simplification of most groundwater food webs, make these organisms very vulnerable to external perturbations coming from the surface (Iannella et al., 2020a; Iannella et al., 2020b). Specifically, persistent stressors such as groundwater abstraction, nutrient enrichment, and introduced contaminants erode population resilience by disrupting the stable physicochemical conditions to which GW taxa are evolutionarily bound (Fiasca et al., 2014; Di Lorenzo et al., 2019). Climate-driven hydrological shifts further compound these pressures by altering recharge regimes and redox conditions, amplifying extinction risk for narrow-ranged taxa (Siegel et al., 2023).

The inadequacy of the conventional area-based conservation applied to surface systems underscores the need for integrated karst-catchment zoning that buffers aquifers from land-use change and maintains ecological connectivity among subterranean refugia (Mammola et al., 2022; Colado et al., 2023; Zagmajster et al., 2014; Mammola et al., 2024). Mounting evidence indicates that strategic, surface-to-subsurface management can secure this hidden biodiversity (Culver & Pipan, 2019; Mammola et al., 2024). Indeed, recently developed tools—ranging from trait-based bioassessment and eDNA metabarcoding to standardized ecotoxicological protocols—now allow early detection of population decline and pollutant sensitivity in subterranean communities, facilitating adaptive management before irreversible change occurs (Fišer et al., 2022). Crucially, successful conservation cases demonstrate that stakeholder-inclusive governance, sustained monitoring, and legal recognition of groundwater’s ecological dimension can arrest biodiversity loss even in tourism-pressured show caves (Deharveng & Bedos, 2018; Wynne et al., 2021).

Conclusion

This study aimed to assess whether caves can serve as reliable proxies for evaluating groundwater copepod biodiversity within European karst systems. By mapping species richness across caves and other karst groundwater habitats, we identified seven biodiversity hotspots and quantified the contribution of cave data to the broader patterns of copepod diversity in the overall karst hotspots.

Our results show that while caves often host a substantial share of the copepod species in karst regions—particularly in areas like the Dinaric Alps, the Cantabrian-Pyrenean ridges, and the Balkans—their representativeness is not uniform. In regions such as the Central Apennines, cave records alone underestimate karst richness, emphasizing the need to include non-cave karst habitats in biodiversity assessments.

These findings support the core hypothesis that caves, though essential and accessible, cannot be the sole basis for karst conservation planning. We highlight the importance of integrated, ecosystem-based approaches that consider the heterogeneity of karst systems and their evolutionary and hydrological complexity.

Future research should focus on addressing current sampling gaps, especially in underexplored karst areas and non-cave environments, and on characterizing cryptic and undescribed taxa. We also recommend the development of spatially explicit conservation strategies that reflect both taxonomic richness and habitat connectivity.

Supplemental Information

Supplemental Information 1 List of species, indicating whether they are found at least once in caves, are exclusive to cave habitats, or shared with other aquifer types

Supplemental Information 2 List of species that triggered the identification of hotspot macroareas, for both karst and cave hotspots

We thank Andrea Massagli (Federazione Speleologica Toscana, Italy) for permission to publish Fig. 1E.

Additional Information and Declarations

Competing Interests

Author Contributions

Data Availability

The authors declare there are no competing interests.

Emma Galmarini conceived and designed the experiments, analyzed the data, prepared figures and/or tables, authored or reviewed drafts of the article, and approved the final draft.

Mattia Di Cicco analyzed the data, authored or reviewed drafts of the article, and approved the final draft.

Barbara Fiasca conceived and designed the experiments, analyzed the data, prepared figures and/or tables, authored or reviewed drafts of the article, and approved the final draft.

Nataša Mori analyzed the data, authored or reviewed drafts of the article, and approved the final draft.

Mattia Iannella analyzed the data, prepared figures and/or tables, and approved the final draft.

Tiziana Di Lorenzo analyzed the data, authored or reviewed drafts of the article, and approved the final draft.

Francesco Cerasoli conceived and designed the experiments, analyzed the data, authored or reviewed drafts of the article, and approved the final draft.

Diana Maria Paola Galassi conceived and designed the experiments, analyzed the data, authored or reviewed drafts of the article, and approved the final draft.

The following information was supplied regarding data availability:

The raw data is available in the Supplemental Files.

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
