# Peer review of "Are caves enough to represent karst groundwater biodiversity? Insights from geospatial analyses applied to European obligate groundwater-dwelling copepods"

_PeerJ, doi:10.7717/peerj.20285_

## Round 0.1 · original submission · Minor Revisions

· Academic Editor

Minor Revisions

Dear author,

Thank you for your submission. We have received two reviews, both of which are highly complimentary of your work. I agree with the reviewers that your manuscript will be a valuable addition to the literature. The reviewers do note some minor issues, however, for your consideration.

I look forward to reading your revised work.

Best,
Anthony

·

Basic reporting

A clear, unambiguous, scientific English language is used throughout the text. Nonetheless, I have entered some corrections/suggestions in the reviewed PDF on some sentences which could be further improved.
The introduction provides a detailed scientific background and nicely contextualises the described research; the aims of the study are clearly described.
The relevant and most recent literature is properly cited; I did not notice any relevant literature missing in the manuscript.
The structure of the manuscript conforms to PeerJ standards.
Figures are few but relevant, of high quality, and well labelled (but see general comments in point 4 for possible improvement) and described.
All raw data were supplied in two supplementary tables (and very conveniently, in xls format, which is a useful tool for those readers who are interested in using it as a regional/national dataset), as well as the permission to publish figure # 1E

Experimental design

The manuscript falls within the scope of the journal
The research questions are clearly listed, and they are very relevant for the study of the biodiversity of groundwater aquatic habitats in general and of karstic systems in particular
The knowledge gaps in the investigated issue are so many, as clearly shown in the introduction and discussed by the authors, that their contribution represents a valuable contribution to fill such gaps.
The analytical approach is very interesting, and based on such a robust and spatially widespread dataset, I have no doubts that the results are supported by the correct data and by a correct analysis.
The methods are properly described, the data source is freely available and properly referred to in the methods section. I have only one request: can you explain how you selected the spatial thresholds (10 km, 5 km, lines 197-199)

Validity of the findings

The research described and discussed in the manuscript is the most complex analysis I am aware of , related to the biodiversity of karstic habitats across Europe, and is the last in a set of previous papers by the same research group, which pioneered in investigating the spatial distribution of GW copepods with a geostatistical approach. The manuscript will be a valuable addition to the literature on this subject, which is relatively scarce, for several reasons properly outlined by the authors. An additional positive note is the listing of the endemism and/or the distribution of the most relevant taxa. As a copepodologist, I find this set of information extremely valuable, as it would allow me to avoid checking all the relevant literature to obtain the description of the distribution range of certain species of interest. All the source data (a very large dataset covering all EU) are freely available, and a link to download them was provided, which is another useful information for other researchers interested in these topics
The final conclusions are well stated, linked to the original research question & limited to the supporting results. I am very partial to the comments related to the need for conservation of these fragile habitats.

Additional comments

I have a few minor comments, which are listed here, ordered as progressive sections/line numbers. Several minor comments, edits, and language-improving suggestions are inserted in the reviewed pdf.
Probably I am not familiar with the format of this journal, but why is the title page with the abstract repeated twice, and with a different order of authors (in the first version, Page 4, main authors: Galassi, Galmarini, Di Cicco, second version, page 6, main authors Galmarini and Di Cicco and Galassi is last author)?
Lines 255-256. I apologize, perhaps I didn’t understand correctly your methods: you have data from only one cave in Sardinia (Bue Marino Cave), and the karst/cave area overlap. However, map 2C reports an area in NW Sardinia (where all the hyporheic records from the EGGop dataset are, which should not have been included in the analysis, as the unconsolidated sediment (hyporheic) habitat should not have been taken into account , which does not correspond to the area of the cave (Eastern coast). Why this discrepancy?
Results section “Karst hotspots in Europe”: perhaps it would help visualizing the results if the macroareas were labelled in figure 2A (and 2B for cave hotspots)
Line 450-451. Centinela extinctions (Wilson, 1999). Nice to use “key concepts” (e.g., the Racovitzan Impediment”), but could you please use an easier sentence (e.g., the disappearance of a species before it could be described) to make reading more fluid?
Line 470. The title of this section, however suggestive, does not represent the contents. Why “hiatus”? Also, the aspect of hydrological connectivity is not discussed.
Line 476-7: The results obtained in the abovementioned studies are surprisingly convergent, to some extent, to the ones emerging in the present analyses. Not so surprisingly, part of the databases are the same. But the mentioned studies used.
Line 481.484. This sentence falls within the discussion on copepod value as indicators and the details of their distribution. Can you please link the following paragraph to it by adding one connecting sentence ( if copepods in caves mark the biodiversity of karst and caves, cave specialists are the relevant component of the diversity). Or, move this sentence before line 500 and link it with the following paragraph.
Line 506-512. I am not sure it is correct to use the term “mesohabitat” as a synonym of microenvironments, in geomorphology, the “meso” and the “micro” spatial (and temporal) scales are by definition different
Line 570: Again (why again? You did not mention Lamoroux before) the statement by Lamoreaux. Which statement? Is the following one a statement? If so, use quotation marks; if not, explain what you mean.
Line 574. The title of this section does not fully represent the contents, which deal mainly with the geological history of the European karst area, not with the faunistic hotspots
Line 620 -621. Regrading this section, I have the, probably, main issue which I kindly ask you to consider: the difference between the karstic and the cave habitats (i.e., the karstic habitats which are not cave/epikarst related) is represented by “karst springs (both intermittent and permanent over time), karst rivers and wells drilled in consolidated rocks across Europe”. Yet, these types of habitats are never specifically mentioned in the discussion. I ask you to elaborate on this; at least for springs, there are numerous relevant publications remarking on their role in sustaining biodiversity and hosting endemic taxa. Spring protection is also a priority for conservation, but it was not mentioned in the last section (implications for conservation), regardless of the fact that your work in the Abruzzi region has detected numerous endemisms from there. Wells are also relevant, as the aquifer they reach can be at different depths and hence at different degrese of connectivity with surface habitats. I ask you to elaborate, even if briefly, on these issues.
Captions to figures
Figure 1, Could you add the actual name of the location? Credits for figure A-D, F?
Is there a caption for the two supplementary tables, and I can’t find it? If there is not, can you please provide it.

Reviewer 2 ·

Basic reporting

The basic reporting section is of satisfactory quality.
The article is written clearly using professional English throughout.
It conforms to professional standards.

Experimental design

The study design portion is well organized.
The methods are described with sufficient detail and information.

Validity of the findings

The validity of the findings is systematically structured.
The conclusions are well stated and linked to the original research question.

Additional comments

The manuscript exhibits excellent writing skills and provides a comprehensive analysis of the biodiversity of karst groundwater, particularly relevant to European groundwater-dwelling copepods. I have only one suggestion: to include a summary table to make this manuscript more appropriately informative.
The contents in Table S1 that are cited in Line 245 are intriguing. The authors include this table in the supplementary file but not in the results (maybe because of the large amounts of information). I would suggest adjusting the contents by putting the numbers of species in each genus instead of the long list of each species. Please refer to my suggested contents in Table S1 in the attached file. Please also include the adjusted Table S1 in the results.

Annotated reviews are not available for download in order to protect the identity of reviewers who chose to remain anonymous.

---

## Round 0.2 · Minor Revisions

· Academic Editor

Minor Revisions

Dear authors,

Thank you for submitting your revision and responses. I do not see the need to request further reviews of your work though I do have a few minor additional requests of my own pertaining to your responses:

1. On L209 your comments detail threshold selection. Please provide a succinct version of these explanations in the text so that the reader can understand your rationale.

2. In your response to reviewer 1 you noted that "This area located in NW Sardinia corresponds to karst landscapes of Capo Caccia-Punta Giglio, but the signal is very low, as the reviewer can see in the maps and is related to some copepod species collected from Dasterru di Punta Giglio (Alghero) and the Nettuno Cave. This point is mentioned in line 654. These records do not come from the hyporheic zone; instead, they come from karst areas." Again, could you please provide relevant information earlier in the text - as indicated by the reviewer - just to minimise the potential for misunderstanding.

Many thanks,
Anthony

---

## Round 0.3 · accepted · Accept

· Academic Editor

Accept

My thanks for your prompt resubmission. I am happy to recommend publication of this work and offer my congratulations.